# Exploiting metabolic acidosis in solid cancers using a tumor-agnostic pH-activatable nanoprobe for fluorescence-guided surgery

F. J. Voskuil [1,13], P. J. Steinkamp [2,13], T. Zhao[3], B. van der Vegt [4], M. Koller[2], J. J. Doff[4], Y. Jayalakshmi[3], J. P. Hartung[5], J. Gao [6,7], B. D. Sumer [6✉], M. J. H. Witjes[1], G. M. van Dam [2,8✉] & the SHINE study group*

Cancer cell metabolism leads to a uniquely acidic microenvironment in solid tumors, but exploiting the labile extracellular pH differences between cancer and normal tissues for clinical use has been challenging. Here we describe the clinical translation of ONM-100, a nanoparticle-based fluorescent imaging agent. This is comprised of an ultra-pH sensitive amphiphilic polymer, conjugated with indocyanine green, which rapidly and irreversibly dissociates to fluoresce in the acidic extracellular tumor microenvironment due to the mechanism of nanoscale macromolecular cooperativity. Primary outcomes were safety, pharmacokinetics and imaging feasilibity of ONM-100. Secondary outcomes were to determine a range of safe doses of ONM-100 for intra-operative imaging using commonly used fluorescence camera systems. In this study (Netherlands National Trial Register #7085), we report that ONM-100 was well tolerated, and four solid tumor types could be visualized both in- and ex vivo in thirty subjects. ONM-100 enables detection of tumor-positive resection margins in 9/9 subjects and four additional otherwise missed occult lesions. Consequently, this pH-activatable optical imaging agent may be clinically beneficial in differentiating previously unexploitable narrow physiologic differences.

[1] Department of Oral & Maxillofacial Surgery, University of Groningen, University Medical Center Groningen, Groningen, The Netherlands. [2] Departments of Surgery, Nuclear Medicine and Molecular Imaging, Medical Imaging Center Groningen, University of Groningen, University Medical Center Groningen, Groningen, The Netherlands. [3] OncoNano Medicine Inc., Dallas, TX 75390, USA. [4] Department of Pathology & Medical Biology, University of Groningen, University Medical Center Groningen, Groningen, The Netherlands. [5] JPH Clinical Development, San Diego, CA 92131, USA. [6] Department of Otolaryngology Head and Neck Surgery, Simmons Comprehensive Cancer Center, University of Texas Southwestern Medical Center, Dallas, TX 75390, USA. [7] Department of Pharmacology, University of Texas Southwestern Medical Center, Dallas, TX 75390, USA. [8] AxelaRx/TRACER B.V, Groningen, The Netherlands. [13] These authors contributed equally: F. J. Voskuil, P. J. Steinkamp. *A list of authors and their affiliations appears at the end of the paper. ✉email: baran.sumer@utsouthwestern.edu; g.m.van.dam@umcg.nl

Cancer specific fluorescence-guided imaging has traditionally involved fluorophore-labeled small molecules, antibodies, nanobodies, peptides, or nanoparticles against cell surface receptors[1–9]. Despite promising results, such strategies often lack broad tumor applicability due to the diversity of oncogenotypes and phenotypes[10,11]. In contrast, extracellular cancer acidosis—a ubiquitous consequence of cell proliferation and growth in cancer—could serve as a generic target in a variety of solid tumors[12,13]. While virtually all solid cancers demonstrate low extracellular pH relative to the tightly regulated pH of the normal extracellular compartment, the pH is spatiotemporally variable. The variability in tumor pH is affected by tumor intrinsic factors, such as aerobic and anaerobic glycolysis, hypoxia, angiogenesis, and perfusion, which interact in unpredictable ways with different areas of a given cancer resulting in a pH between 5.8 and 7.4 at different times[14,15].

To exploit the dysregulated tumor milieu, a series of tunable pH-sensitive amphiphilic polymers have been developed that generate a fluorescent output in response to a reduction in pH[16]. In an aqueous solution, the polymers self-assemble into nanometer-sized micelles. Fluorophores, such as indocyanine green (ICG), sequester within the hydrophobic segments of the micellar core leading to homoFRET fluorescence quenching[17]. Cooperative dissociation of the micelles at a tunable pH threshold releases the individual unimers, thereby unquenching the fluorescent dye (Fig. 1). The discrete cooperative response of the polymers to pH is a unique nanoscale phenomenon with two distinguished characteristics for pH sensing[18]. First, the cooperativity, with a Hill's coefficient as high as 51, makes fluorescence activation a thresholded maximal event: fluorescence is either completely off or completely on above and below a sharply demarcated pH point. Second, the activation is irreversible due to the capture of dissociated unimers by the proteins in the tumor microenvironment which prevent reformation of the micelles that would quench the fluorescence. When progressive fluorescence activation in zones of acidosis is integrated over time, a stable, discrete fluorescent labeling of tumors is achieved[19].

Human patients vary widely with respect to body habitus and underlying metabolic and nutritional status, while their cancers vary by type, size, and mutational status[20]. These extrinsic and intrinsic tumor variables compound accurate tumor pH detection. Despite Otto Warburg first describing deregulated cancer metabolism over a century ago, it has not been utilized clinically. We therefore hypothesized that the nanoscale cooperativity of the pH-responsive polymers could be leveraged for fluorescence-guided tumor identification and that the irreversible binary activation could provide novel information not obtainable through routine surgical standards of care.

In this clinical study, ONM-100, an intravenously administered imaging agent based on pH-sensitive polymers, was investigated in human patients. The results presented below demonstrate ONM-100's ability to detect, otherwise missed, tumor-positive surgical margins and occult disease. All four of the investigated solid tumor types were visualized by tumor-agnostic fluorescence. The potential of ONM-100 in clinical decision making during and post-surgery needs to be confirmed by an additional Phase II clinical study.

## Results

**Summary of safety and pharmacokinetic evaluations of ONM-100.** The primary outcomes of this Phase 1 study were the safety, pharmacokinetics, and imaging feasibility of a single intravenous dose of ONM-100 administered to patients with solid cancer 24 ± 8 h prior to undergoing standard surgery. The study procedures are shown in Fig. 2. Thirty subjects with four different solid tumor types (head and neck squamous cell carcinoma [HNSCC], n = 13, breast cancer [BC], n = 11, esophageal cancer [EC], n = 3, and colorectal cancer [CRC]), n = 3 were enrolled.

Eight subjects with HNSCC and seven with BC were included in the dose-escalating Phase 1a study (secondary outcome). To evaluate the tumor-agnostic imaging feasibility further with ONM-100, 15 additional subjects with four different tumor types (HNSCC, BC, EC, or CRC) were dosed in Phase 1b with an optimal dose determined in Phase 1a (Table 1 and Supplementary Table 1). All 30 subjects completed the study. ONM-100 was well-tolerated by all 30 subjects with no dose limiting toxicities or drug-related serious adverse events (SAEs). Four possible drug-related Common Terminology

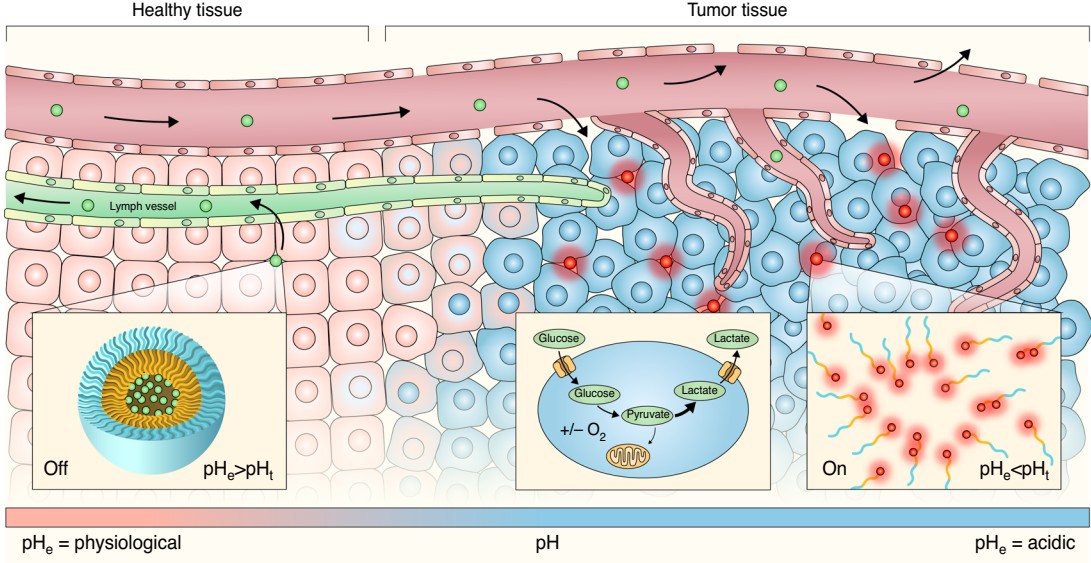

**Fig. 1 Mechanism of action.** A tumor microenvironment turns acidic when the tissue transforms from pre-malignant cells into invasive cancer. The ONM-100 nanoparticles extravasate due to the enhanced permeability of the tumor vasculature and are then retained due to the poor lymph drainage in the tumor tissue. This leads to ONM-100 accumulation in the acidic extracellular matrix causing the pH activated fluorescence to switch from the "off" (green) to the "on" (red) state. $pH_e$: extracellular pH, $pH_t$: threshold pH.

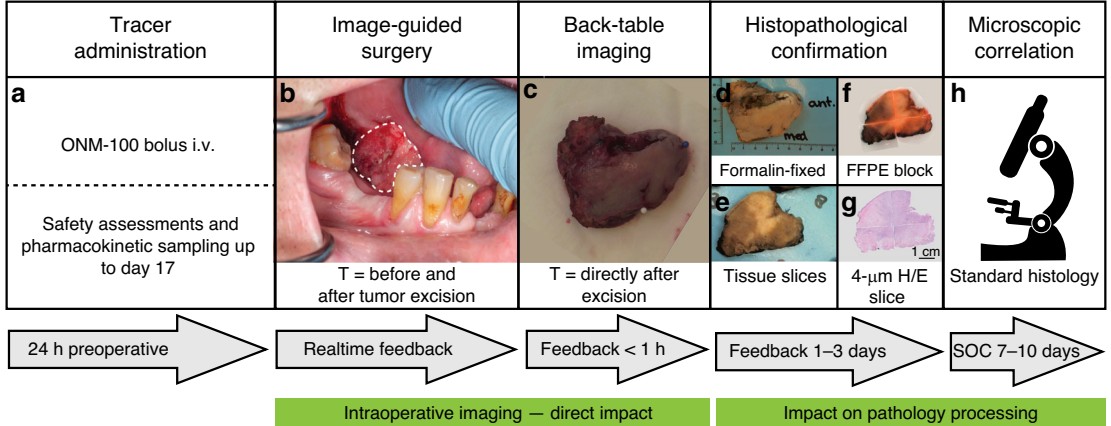

**Fig. 2 Study design.** Intravenous administration of ONM-100 was performed ~24 h (±8 h) prior to surgery. Ten days of safety assessments (laboratory, pharmacokinetics, ECGs) followed, adverse events were monitored up to day 17 (**a**). During surgery, intraoperative images were obtained prior to incision and after excision of the surgical cavity (**b**). Immediately after excision the specimen was imaged for the presence of a positive surgical margin (**c**). Fluorescence images were obtained during all the standard pathology processing phases (**d**, **e**), and the H/E slices were correlated with the standard histopathology slices (**f**–**h**). i.v. intravenous, ECG electrocardiogram, H/E hematoxylin eosin, SOC standard of care.

**Table 1 Study characteristics.**

| Tables | All N = 30 | 0.1 mg per kg N = 3 | 0.3 mg per kg N = 3 | 0.5 mg per kg N = 3 | 0.8 mg per kg N = 3 | 1.2 mg per kg N = 18 |
|---|---|---|---|---|---|---|
| *Patient characteristics* | | | | | | |
| Age, mean (range) | 63 (35–85) | 68 (46–80) | 53 (35–63) | 57 (50–69) | 61 (53–78) | 65 (34–85) |
| Males, number (percentage) | 8 (27%) | 1 (33%) | 0 (0%) | 1 (33%) | 0 (0%) | 6 (33%) |
| Weight, mean (range) | 80 (47–113) | 62 (53–69) | 89 (71–113) | 88 (52–119) | 87 (73–94) | 79 (47–110) |
| *Safety data* | | | | | | |
| AE Grade I | 0 | 0 | 3 | 0 | 0 | 0 |
| AE > Grade I | 0 | 0 | 0 | 0 | 0 | 0 |
| *Tumor type* | | | | | | |
| BC | | | | | | |
| Invasive carcicoma NST | 7 | 0 | 2 | 1 | 2 | 2 |
| Carcinoma with medullary characteristics | 2 | 0 | 1 | 0 | 0 | 1[a] |
| Mucinous carcinoma | 1 | 0 | 0 | 0 | 0 | 1 |
| Lobular carcinoma | 1 | 0 | 0 | 0 | 0 | 1 |
| Invasive micropapillary carcinoma | 1 | 0 | 0 | 0 | 0 | 1[a] |
| HNC | | | | | | |
| HNSCC | 13 | 3 | 0 | 2 | 1 | 7 |
| EC | | | | | | |
| Intestinal adenocarinoma | 1 | 0 | 0 | 0 | 0 | 1 |
| Mixed adenoneuroendocrine carcinoma | 1 | 0 | 0 | 0 | 0 | 1 |
| No viable tumor (after neoadjuvant Tx) | 1 | 0 | 0 | 0 | 0 | 1 |
| CRC | | | | | | |
| Intestinal adenocarcinoma | 3 | 0 | 0 | 0 | 0 | 3 |
| *Tumor size* | | | | | | |
| Max diameter, mean (±SD) | 2.4 (±1.6) | 1.9 (±0.4) | 2.2 (±0.8) | 3.6 (±1.0) | 3.6 (±2.2) | 2.1 (±1.6) |
| *Surgical resection margins* | | | | | | |
| Tumor-negative | 18 | 11 | 2 | 2 | 2 | 1 |
| Tumor-positive | 9 | 4 | 1 | 1 | 1 | 2 |

Surgical resection margins are displayed according to the US Guidelines per tumor type. Adverse events are defined as possible imaging agent related adverse events. Source data are provided as a Source data file.
*BC* breast cancer, *HNC* head and neck cancer, *HNSCC* head and neck squamous cell cancer, *EC* esophageal cancer, *CRC* colorectal cancer, *SD* standard deviation, *Tx* treatment, *NST* no special type, *AE* adverse event.
[a]Two primary tumors from the same subject.

Criteria for Adverse Events (CTCAE) grade 1 adverse events (AEs) were observed in the 0.3 mg per kg cohort (Table 2). General blood counts, determined up to day 10, showed no imaging agent related aberrations. The plasma exposure (C10m and AUC 0–24 h) of ONM-100 was linearly correlated with the dose, with an $R^2$ of 0.95 and 0.98, respectively (Supplementary Fig. 1). The mean terminal-phase half-life in the 1.2 mg per kg cohort was 44.5 with a standard deviation of 15.8 h. No differences in pharmacokinetics were observed among the different tumor types (Supplementary Fig. 2).

**Table 2 Adverse events.**

| Tumor type | Dose cohort | Adverse event | Grade | Intervention | Resolved |
|---|---|---|---|---|---|
| BC | 0.3 mg per kg | Headache | I | No | Yes |
| BC | 0.3 mg per kg | Pain left cheek | I | No | Yes |
| BC | 0.3 mg per kg | Flush | I | No | Yes |
| BC | 0.3 mg per kg | Dizziness | I | No | Yes |

Adverse events that occurred during the course of the study which were possibly related to imaging agent administration and scored according to the Common Terminology Criteria for Adverse Events (CTCAE) v4.03: June 14, 2010.
BC breast cancer.

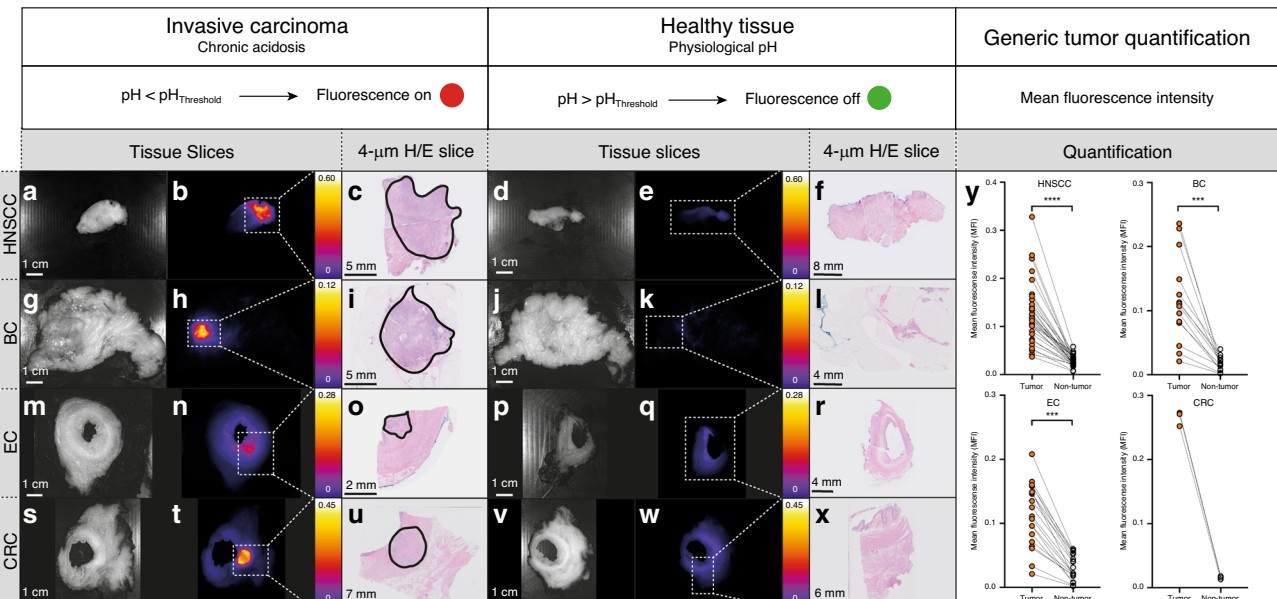

**Fig. 3 Fluorescence images of different tumor tissue slices.** Head and neck squamous cell cancer of the tongue (**a**–**f**); breast cancer (**g**–**l**); esophageal cancer (**m**–**r**); colorectal cancer (**s**–**x**). The tumor is delineated as a solid black line in the H/E slices (**c**, **i**, **o**, **u**). The mean fluorescence intensity (MFI) of the tumor tissue and the non-tumor tissue slices, per tumor type, is depicted (**y**). The dots represent the MFI of single tissue slices (≈3 per subject) from the 1.2 mg per kg cohort. HNSCC, 7 subjects, $P < 0.0001$, BC, 5 subjects, $P = 0.0001$, EC, 3 subjects, $P = 0.0010$, and Wilcoxon test, two-sided. CRC, 3 subjects, no statistics performed due to the availability of only three data points. HNSCC head and neck squamous cell cancer, BC breast cancer, EC esophageal cancer, CRC colorectal cancer, H/E hematoxylin eosin. ***$P ≤ 0.001$; ****$P ≤ 0.0001$. Source data are provided as a Source data file.

**Table 3 Contingency table for fluorescence-guided surgical margin assessment.**

| | Tumor-positive resection margin | Tumor-negative resection margin |
|---|---|---|
| Fluorescence positive | 9 | 5 |
| Fluorescence negative | 0 | 10 |
| Total | 9 | 15 |

Intraoperative assessment of the surgical margin during fluorescence-guided surgery was either done by intraoperative fluorescence imaging of the surgical cavity or fluorescence imaging of the excised specimen at the "Back-Table".

**Summary of fluorescence-imaging results**. Viable tumors were confirmed in a total of 29 of the 30 enrolled subjects by histology. In each case, a sharp demarcated fluorescent signal was visible irrespective of the tumor type or dose (Fig. 3) based on the ex vivo standard fluorescence workflow analysis described previously[21] (Fig. 2). We confirmed the tumor specific activation of ONM-100 ex vivo by administering it topically on tissue sections of a freshly frozen HNSCC specimen (Supplementary Fig. 3). Moreover, no fluorescence was visible in the blood samples prior to acidification (Supplementary Fig. 4).

Thirteen HNSCC, the five superficially seated BC tumors and two CRC tumors were visualized in situ in real time by ONM-100 fluorescence during surgery. All the tumor-positive surgical

margins (9 out of 9) that were undetected during standard of care (SOC) surgery were visualized during intraoperative fluorescence imaging and correlated with the final histopathological assessment, yielding 100% sensitivity and no false negatives (Table 3 and Fig. 4). In addition, ONM-100 fluorescence was detected in five additional occult lesions (in 1 HNSCC and 4 BC subjects) otherwise missed by the SOC surgery or pathological analysis. The histologically proven peritoneal metastases (PM) of the two CRC subjects were fluorescent both in- and ex vivo, whereas their nonmalignant excised lesions did not fluoresce (Supplementary Figs. 5 and 6 and Supplementary Movie 1–4).

The tumor-agnostic nature, diagnostic accuracy, and potential clinical utility of ONM-100 are further described below. Three of

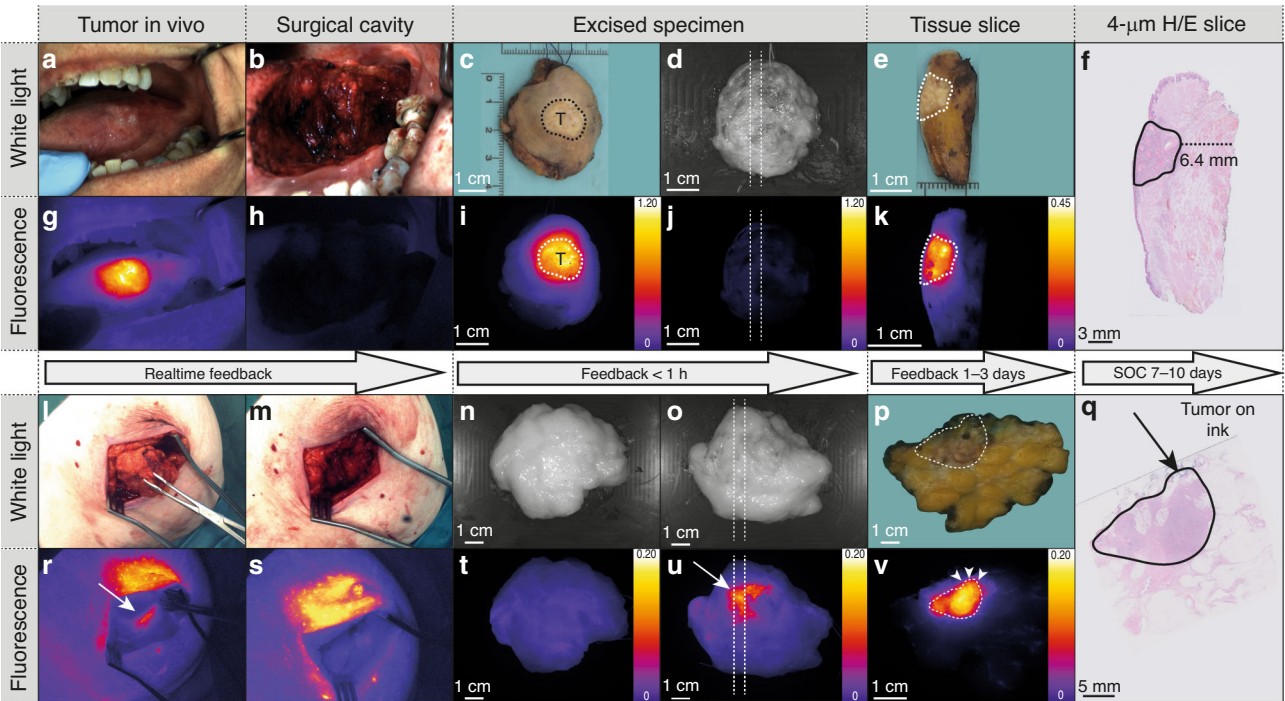

**Fig. 4 Fluorescence-guided assessment of surgical margins.** Representative example of a head and neck squamous cell carcinoma of the tongue from a subject with a negative surgical margin. In- and ex vivo visualization of fluorescence in the tumor (**a**, **c**, **g**, **i**) with no fluorescent signal in the surgical cavity or at the surgical resection margin (**b**, **h**, **d**, **j**). Correlation of fluorescent signals on a tissue slice with the histology (**e**, **k**, **f**) with a tumor-negative surgical margin of 6.4 mm. Representative example of breast cancer surgery (i.e., a lumpectomy) with a tumor-positive surgical margin (**l**, **m**, **n**, **o**). Fluorescence is detected at the ventral surgical resection margin both in vivo and immediately after excision (**r**, **s**, **t**, **u**) which corresponds with the fluorescence localization on the tissue slice (**p**, **v**) and the final histopathology (**q**). The tumor is delineated as a solid black line on the H/E slices (**f**, **q**). H/E hematoxylin eosin, SOC standard of care.

the BC subjects (27%) had a tumor-positive surgical margin, all of which were detected by fluorescence imaging. Eight BC subjects had a proven histopathological tumor-negative surgical margin of which six were assessed correctly by fluorescence imaging, corresponding to a sensitivity of 100% and a specificity of 75% for ONM-100 in this sample size (Supplementary Fig. 7 and Supplementary Table 2). Of the two BC subjects with a false-positive fluorescence-imaging result, one subject had a histopathological tumor-negative margin as defined by the ASTRO/SSO guidelines. However, it contained a ductal carcinoma in situ (DCIS), an entity with cancer cells within the wall of the ductuli and, according to international guidelines, might require additional surgery, underscoring the clinical utility of detecting this lesion. Hence, it can be debated if this represented a true false-positive sample. In the other BC subjects with a false-positive fluorescence-imaging result, the fascia of the larger pectoral muscle showed a homogenous higher fluorescent signal, as also described previously[21]. Regarding the HNSCC cohort, a total of six subjects (46%) had a histopathologically confirmed tumor-positive surgical margin, all of which were detected intraoperatively by fluorescence. A tumor-negative surgical margin was confirmed histopathologically in the remaining seven subjects (54%). Three false-positive fluorescence margins were detected, which did not contain tumor on final histopathological examination (Supplementary Fig. 5), resulting in a sensitivity of 100% and a specificity of 57% among the HNSCC subjects (Supplementary Fig. 7 and Supplementary Table 1).

The ability to detect occult disease such as satellite metastases and second primary tumors, which are missed by SOC procedures, exemplifies the potential clinical utility of ONM-100. This was an ad hoc outcome parameter of the study. A satellite metastasis in one of the HNSCC subjects was undetected

by SOC surgery but was detected in the surgical cavity by fluorescence imaging (Supplementary Fig. 5). As discussed above, the DCIS was detected by ONM-100 fluorescence in the surgical cavity and back-table in a specimen of a BC subject and was confirmed by histopathology. Moreover, fluorescence imaging during histopathological processing detected three additional otherwise missed cancers in three BC subjects. Of these, fluorescence imaging enabled the detection of additional satellite BC metastasis in the surgical specimens of two subjects and a second primary tumor lesion (triple negative BC) in the third subject (Supplementary Fig. 5).

One deeper-seated intraluminal rectal, two intraluminal esophageal, and six deeper-seated breast tumors could not be visualized intraoperatively due to the near-infrared (NIR) imaging technology. Notably, however, none of the intraluminal or deep-seated tumors had a tumor-positive margin on final histopathology. Importantly, the fluorescence-imaging procedure did not interfere with the SOC of any of the tested subjects and, generally, the surgical procedures were only prolonged by a maximum of 10 min.

**Fluorescence quantification and tumor-to-background ratio.** Ex vivo workflow analysis, to further validate the intraoperative findings, showed that the tumor tissue of all the subjects with histopathologically proven viable tumor tissue showed a higher fluorescence signal intensity with sharp morphological delineations on the tissue slices compared with normal tissue, irrespective of tumor type and dose cohort (Fig. 3, panel y and Supplementary Fig. 6). The tumor tissue's mean fluorescence intensity (MFI) increased with dose (Fig. 5, panel a). In all cohorts, the tumor MFIs were significantly higher than that of

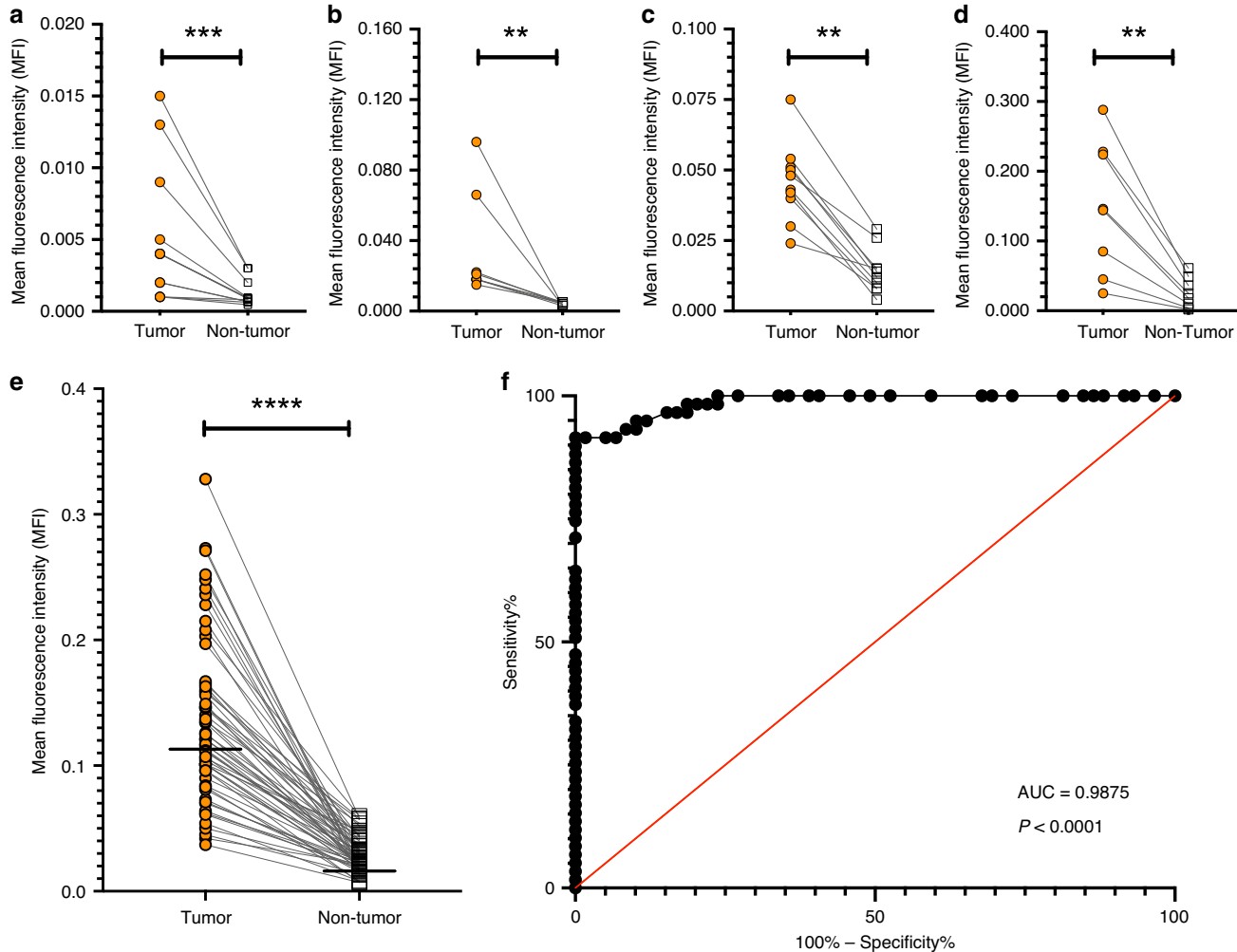

**Fig. 5 Dose-independent mean fluorescence intensity separation between tumor tissue and non-tumor tissue.** Tumor and non-tumor tissue mean fluorescence intensities (MFI) from the 0.1 mg per kg cohort, $P = 0.0005$ (**a**); 0.3 mg per kg cohort, $P = 0.0078$ (**b**); 0.5 mg per kg cohort, $P = 0.0020$ (**c**); 0.8 mg per kg cohort, $P = 0.0078$ (**d**); and 1.2 mg per kg cohort, $P < 0.0001$, Wilcoxon test, two-sided (**e**). The dots represent the MFI of single tissue slices. The receiver-operator characteristics curve is based on the calculated MFI of the tumor and normal tissues from the 1.2 mg per kg dose cohort, $P <$ 0.0001; area under the curve 0.9875, $n = 59$, with a confidence interval of 95% using Wilson/Brown method (**f**). ROC receiver operators curve, AUC area under the curve. $**P \leq 0.01$; $***P \leq 0.001$; $****P \leq 0.0001$. Source data are provided as a Source data file.

non-tumor tissue (Fig. 5). The median tumor-to-background ratio (TBR) of all the tissue slices ($n = 97$ from 26 subjects) was 4.5 with an interquartile range (IQR) of 3.1 (Fig. 5, panel a). The optimal dose for tumor detection and sensitivity according to the Phase 1b studies was 1.2 mg per kg (TBR 4.5, IQR 3.0) and the MFI of that dose group's ($n = 15$) tumor tissue was significantly higher compared with normal tissue in each of the available tissue slices ($n = 59$, $P < 0.0001$, Wilcoxon test). A receiver-operator characteristic (ROC) curve analysis of these tissue slices showed an area under the curve (AUC) of 0.9875, $P < 0.0001$ (Fig. 5, panel g).

An in vivo TBR was calculated for the mucosal tumors (HNSCC) which were directly exposed to the surface; deeper-seated tumors gave less reliable results due to overlaying tissue and differences in absorption and the scattering properties of the tissue. The median in vivo TBR of the HNSCC subjects in the 1.2 mg per kg group was 2.6 with an interquartile range of 1.4 (representative examples shown in Supplementary Fig. 8).

## Discussion
The in and ex vivo data from this fluorescence-imaging study indicate that the low pH resulting from tumor acidosis can be

exploited as a generic biomarker for cancer in patients with four different solid tumor types including head and neck squamous cell carcinoma, breast cancer, esophageal cancer, and colorectal cancer. The pH-sensitive fluorescent imaging agent ONM-100 was well-tolerated and was activated by tumor acidosis, delineating tumors from benign tissues. The initial clinical findings from 30 subjects provide information about occult cancer which would otherwise not be obtained by the standard of care (i.e., visual inspection and palpation alone), which was illustrated by the intraoperative detection of all the tumor-positive surgical margins (9 out of 9), a DCIS and a satellite cancer in a head and neck cancer subject as well as the ex vivo detection of three additional satellite lesions and second primaries in the pathology specimens.

This Phase I study is the first example of a systemically administered agent that displays nanoscale cooperativity which overcomes metabolic and phenotypic variability between different patients and tumors. Most importantly, and in contrast to recently studied fluorescence-imaging agents[1–8], there was no overlap between tumor and background fluorescence for any given subject.

Two features of the reported data are unique. First, while tumor acidosis was first described almost 100 years ago[22], clinical

use of this physiologic characteristic of cancer has been hampered by the spatiotemporal variability of pH in tumors and the wide array of factors that can impact tumor acidity[11]. We were able to utilize tumor acidosis in a clinical setting, in diverse cancer types. Second, although the unique features of nanoscale cooperativity and non-covalent self-assembly have been long recognized, they have not been clinically exploited by a systemically administered agent. The salient features of nanoscale cooperativity, based on macromolecular non-covalent interactions, include weak and polyvalent interactions driving self-assembly, the creation of larger structures that cannot be easily synthesized by covalent chemistry, faster responses to environmental stimuli due to the reduced energy barriers, as occurs with covalent bonds, and most importantly, emergent properties where the system behavior cannot be predicted by studying the individual components in isolation[23].

Our ex vivo spraying experiments give additional proof of the specificity of the imaging agent, as these results indicate that no other mechanism, such as the enhanced permeability and retention (EPR) effect, other than pH is responsible for fluorescent activation. This was illustrated by the fact that tumor specific fluorescence was observed after topical administration of ONM-100, emphasizing that the EPR effect in vivo is responsible for the selective accumulation of micelles in the tumor, but the pH of the tumor is responsible for the opening of the micelle and the activation of the fluorescence signal. Moreover, mechanical destruction of the agent can be excluded since no fluorescence was observed in the plasma samples of patients without acidification.

A potential drawback might be the activation of ONM-100 by other mechanisms associated with a lowered pH, such as in inflammatory tissue[24]. Some tumors present with a peri-tumoral inflammatory rim as part of the defense mechanism of the innate immune system to the invading tumor cells[25]. In this study, we did not observe this phenomenon in this small series of specimens. However, we believe that, if this occurs, it has a minimal impact, and it might actually provide a fluorescent rim surrounding the actual tumor-positive margin, and thus serve the clinical applicability in fluorescence-guided surgery.

Biomarkers like HIF-1-α[26], CA-IX[27], and VEGF-A[11], among others, all play a significant role in the development and maintenance of tumor acidosis. As activation of ONM-100 relies on tumor acidosis in general, and is not dependent on one specific biomarker, immunohistochemical staining would only give limited information on the molecular mechanism of fluorescence activation. It would be beneficial to correlate the in vivo data with the microscopic distribution of the fluorescence data. Unfortunately, this was not feasible in the current clinical trial since ONM-100 is not membrane-bound and the fluorescence was washed out during the standard of care processing of tissues. Invasive pH measurements to correlate the localization of fluorescence with pH within and surrounding the tumor was deemed neither feasible nor safe in this Phase 1 clinical trial due to the following reason: undertaking pinpoint invasive pH measurements in the tumor can influence the integrity of the surgical specimen, potentially hampering standard of care histopathological evaluation. A possible remedy for this, in future studies, could be pH measurements with, for example, chemical exchange saturation transfer magnetic resonance imaging (CEST-MRI)[28] in patients who receive ONM-100.

The timing of the administration of ONM-100 was kept consistent, to 24 h prior to surgery in this Phase 1 feasibility and imaging study, to prevent heterogeneity in the imaging data and was chosen based on pre-clinical data[17]. A different dose-to-imaging interval is being investigated in an ongoing Phase 2 study of ONM-100, since administration only a few hours prior to

surgery might be preferable in terms of clinical implementation. In addition, higher dosages could be investigated in subsequent studies, since this might further improve TBRs. Moreover, our results indicate that ONM-100 could be utilized in a broad variety of other solid tumor types.

Clinical success can be improved when surgeons are provided with a more accurate and unambiguous delineation of the cancer location in addition to the extensive information on the locations of the tumors from conventional ultrasound, CT, PET, or MR imaging. The ability of an optical imaging output to improve surgical outcomes is predicated on delivering information the surgeon does not have from pre-operative imaging and intraoperative inspection[29] by vision and palpation alone. The combination of tumor acidosis as a general phenomenon and the cooperativity of ONM-100 has the potential to adequately improve surgical guidance in a variety of tumors using fluorescence. The broader implication is that nanoscale macromolecular cooperativity can exploit a labile physiologic parameter for clinical purposes. Other biologic parameters (e.g., hypoxia, redox potential) which were previously unexploitable due to unpredictable variability may also be amenable to clinical targeting based on this chemical principle, representing a new therapeutic paradigm.

Nanoscale macromolecular cooperativity, which responds to pH changes from cancer acidosis, is demonstrated to be safe and clinical utility of the ONM-100 imaging agent for intraoperative and ex vivo detection of cancer is shown in this study. The results of our clinical study serve as a proof of principle that physiologic parameters such as pH, which were previously inaccessible therapy targets, may become clinically relevant through the application of macromolecular cooperativity.

## Methods

**GMP synthesis of the pH-activatable micelles**. Clinical grade pH-activatable ONM-100 was released from the Good Manufacturing Practice (GMP) facility of Bioserv Corporation, San Diego, USA, under FDA regulated conditions. The product was shipped to PCI Pharma Services, Bridgend, UK, for distribution to the Hospital Pharmacy Trial Bureau of the University Medical Center Groningen (UMCG), The Netherlands. A detailed description of the production process was published by a previous study[17]. Briefly, ONM-100 consists of polymeric micelles labeled with IndoCyanine Green (ICG). Chemically, the ONM-100 drug substance comprises a diblock copolymer of polyethyleneglycol (PEG) (~113 repeating units) and a poly(methyl methacrylate) derivative covalently conjugated to functionalized ICG as the fluorophore. The ICG content was determined by a qualified method from its molecular weight of 37.5 ± 12.5 kD. Vials containing 9 mg ONM-100 drug substance formulated in sterile water for injection with 10% (v/w) trehalose were used to prepare infusions at a concentration of 3 mg/ml. After the final product was released by the certified qualified person (QP) at the UMCG Hospital Pharmacy facility, the imaging agent was intravenously administered to the subjects.

**Subject population**. The subjects included in this first-in-human clinical study had histologically proven primary or recurrent HNSCC, BC, EC, or CRC. Subjects were eligible if they were 18 years of age or older, and their hematological status was acceptable as was their kidney and liver function. The subjects had to abstain from alcohol intake during the study period (from 24 h before to 17 days after imaging agent administration). Any subjects receiving neoadjuvant therapy prior to surgery were excluded from Phase 1a. Other exclusion criteria were an inability to give informed consent, participation in another clinical trial with an investigational product, inadequately controlled hypertension, history of allergic or infusion reaction to iodine, iodine-based contrast, shellfish, or ICG, those receiving potentially highly hepatotoxic medication, pregnancy, and subjects with magnesium, potassium, and calcium lower than the lower normal limit.

**Clinical trial design**. This dose-finding Phase 1 study was performed in two medical centers and enrolled 30 subjects from March 2018 until December 2018. The study, conducted in both the University Medical Center Groningen (UMCG, Groningen, The Netherlands) and the Martini Hospital Groningen (MZH, Groningen, The Netherlands) was approved by the Institutional Review Board (IRB) of the UMCG (METc Number 2017/580).

The study was conducted according to the Dutch Act on Medical Research involving Human Subjects (WMO) and to the principles of the Declaration of Helsinki (adapted version Fortaleza, Brazil, 2013). All the subjects were identified

by the respective multi-disciplinary tumor boards of the participating hospitals. After a pre-screening, the eligible subjects were orally informed and received written information about the study. All the participants gave written informed consent before the start of any related study procedure. An independent medical monitor was assigned to review the screening safety assessments prior to enrollment. Serious adverse events, if present, were immediately reported to the IRB, the medical monitor and the Dutch central committee on research involving human subjects (CCMO) and were followed up until resolved or a stable medical situation was achieved. The trial was registered at the Netherlands National Trial Register (NTR number 7085) and within the European Clinical Trials Database (EudraCT 2017-003543-38). Primary outcomes were safety, pharmacokinetics, and imaging feasibility of ONM-100. Secondary outcomes were to determine a range of safe doses of ONM-100 for intraoperative imaging using commonly used fluorescence camera systems. The complete study was divided in two phases, Phase 1a and Phase 1b, respectively. We adhered to the FDA guidelines (Guidance for Industry, Developing Medical Imaging Drug and Biological Products, Part 2 Clinical Indications) when designing the Phase 1a dose-finding protocol, followed by setting a dose expansion cohort (15 subjects, Phase 1b). Five different doses were used in an escalating/de-escalating scheme (i.e., 0.3 mg per kg, 0.5 mg per kg, 0.8 mg per kg, 0.1 mg per kg, and 1.2 mg per kg) and administered to three subjects each. In Phase 1a, only subjects with histologically proven HNSCC and BC were included to minimize tumor heterogeneity and to collect information about the potential tumor-agnostic characteristics of ONM-100. After dosing each cohort, a dose escalation meeting was held with the principal investigators, sub-investigators, and the sponsor's medical monitor to make sure the escalation to a higher dose was safe. The most optimally performing dose was 1.2 mg per kg in Phase 1a, based on tumor visualization and safety data. The additional 15 subjects included in Phase 1b were studied to confirm the safety and to evaluate the tumor-agnostic properties of ONM-100 imaging in additional solid tumor types (EC, CRC) to HNSCC and BC. In both phases, the subjects received a single dose of ONM-100 24 ± 8 h prior to surgery. All doses were injected intravenously at 15 mg per min. After the injection, the infusion line was flushed with 5 ml sterile water.

**Safety measurements**. A primary outcome of our study was to measure the safety of ONM-100. The subjects underwent a medical screening procedure before enrollment in the study, consisting of vital signs measurements, a physical examination, a standard 12-lead electrocardiogram (ECG), and laboratory tests (including a serum pregnancy test of women of childbearing potential). Once enrolled, an ECG was performed 1 h and 10 days after the ONM-100 infusion (the latter only in Phase 1a). Lab tests were performed before and after administration (day 1, 2, 3, and 10). The vital signs and physical examinations were measured before and after imaging agent injection (5 min, 1 h, day 2, 3, and 10). The subjects were asked about signs and symptoms before and after the injection (5 min, 1 h, 3 h, 8 h, day 2, 3, 10, and 17). Standard postoperative checkups were arranged for within 2 weeks after surgery. During these visits, wound healing and adverse events were monitored as well. Adverse event assessment was performed according to the National Cancer Institute CTCAE version 4.0.

**Pharmacokinetic assessments**. An additional primary outcome was to measure the pharmacokinetics of ONM-100. Blood samples were collected for pharmacokinetic (PK) analysis prior to, and after 10 min, 30 min, 1 h, 3 h, 8 h, 24 h, 48 h, 72 h, and 240 h of intravenous administration of ONM-100. The blood was collected in 4 mL BD $K_2$EDTA vacutainers and stored directly on ice. The samples were then centrifuged at $1500 \times g$ for 10 min and divided into three vials under cold conditions. The vials were stored in a −80 degrees Celsius freezer in the UMCG and then transported on dry ice to Intertek Pharmaceutical Services (San Diego, California, USA).

The ONM-100 plasma concentrations were determined using a validated direct fluorescence reader assay (Intertek Pharmaceutical Services, San Diego, California, USA) and the PK analysis was performed by Pacific BioDevelopment (Davis, California, USA). Plasma concentration versus time profiles were generated for each subject. The PK parameters were estimated using Phoenix WInNonlin (version 8.0). The estimated parameters were C10m, Cmax, Tmax, AUClast, AUCall, and AUC 0-24hr. Values below the level of quantitation (<10 μg/ml, BQL) were set to 0.

The linear trapezoid method was used for the estimation of the area under the plasma concentration versus time curves from dosing to the last time point with a measurable concentration (AUClast). The last three or more time points were used to estimate the elimination rate constant (λz) which was used to estimate the terminal-phase half-life (T ½).

**Surgical procedure (standard of care)**. All the subjects underwent surgical removal according to the standard surgery protocols of both hospitals for each respective tumor type. Dependent on the tumor type and/or stage, a sentinel lymph node biopsy (lymph node mapping using $^{99m}$Technetium and perioperative detection using a gamma-probe) or a lymph node dissection was performed on some of the subjects. Based on prior experience using fluorescence-guided surgery at the UMCG and MZH, there was minimal interference with the standard of care. As described earlier, the use of methylene blue was avoided and the use of fluorescent skin markers and (green) fluorescent sterile drapes was minimized[21] due to the potential interference with ONM-100.

**Intraoperative fluorescence-imaging devices**. A secondary outcome of our study was to investigate different imaging systems. Two open-surgery intraoperative fluorescent camera systems were used in this study to detect ONM-100, namely the Explorer Air® (SurgVision B.V., Groningen, The Netherlands) and the SPY Elite® (Stryker, Kalamazoo, MI, USA). When performing minimally invasive surgery (e.g., robot assisted esophagectomy or diagnostic laparoscopic surgery for peritoneal metastasis), clinically available near-infrared (NIR) imaging systems were used namely the Olympus NIR Laparoscope (Olympus, Sjinjuku, Tokyo, Japan) and the Intuitive Da Vinci Firefly robot NIR laparoscope (Intuitive Surgical, Sunnyvale, CA, USA).

The Explorer Air®, which provides real-time simultaneous fluorescence and white light (color) images, is currently undergoing CE-marking in Europe and is only available in the European and US markets for experimental use. Fluorescence is excited by NIR light emitting diodes (LEDs) with an excitation peak of 760 nm. Filtered white light is used to illuminate the color images. A software user interface enables the user to control the camera settings and to display the color and fluorescence images. The output is that both still images and movies can be recorded and stored in a TIFF format. Although the excitation peak is not optimized for ICG detection, ICG is efficiently detected due to overlapping light spectra. The working distance of the imaging system was set at 20 cm above the surgical field. All images were obtained with a fluorescence exposure time of 100 ms and a 100 gain. If oversaturation occurred, the exposure time was lowered to 50 or 25 ms but if it persisted, the gain was lowered to 10.

The SPY Elite provides real-time fluorescence and white light imaging. The system is CE-marked for ICG detection and is commercially available. NIR light from the illumination module in the imaging console is transmitted to the imaging head via fiber-optic cables. The SPY Elite has an excitation peak of 805 nm. A software user interface enables the user to control the camera settings and to capture white light images and fluorescence movies. The working distance of the imaging system was set at 30 cm above the surgical field. All images were obtained with a frame rate of 7.5 per second. White light images and fluorescence movies were recorded and stored from the SPY Elite in AVI/PNG format.

The Da Vinci Firefly and the Olympus NIR Laparoscope provide real-time fluorescence and white light images. Both systems, which have an excitation peak of 805 nm, are CE-marked for ICG detection and are commercially available. A software user interface is provided. The working distance of the imaging system was set at 2–20 cm above the surgical field. As described earlier, all the intraoperative camera systems were calibrated using a calibration disk[21] (CalibrationDisk©, SurgVision BV, The Netherlands). The ICG used in the CalibrationDisk© was stabilized by dissolving it in methanol and it was then diluted to different concentrations ranging from 0.005 mg per ml to 1.5 mg per ml to check whether the systems could detect low and high fluorescent signals and as a diagnostic test of the systems.

**Fluorescence-imaging systems for ex vivo imaging**. To further validate the intraoperative imaging feasibility as a primary outcome, we used the the closed-field macroscopic fluorescence PEARL-trilogy® imaging device (Li-COR BioSciences Inc., Lincoln, NE, USA), which is designed for ex vivo fluorescence imaging. A charge-coupled device (CCD) camera detects fluorescence in the NIR wavelength with a peak emission at 785 nm. The 11.2 cm × 8.4 cm field of view and the focus point can be adjusted based on specimen height. The same resolution setting was used (85 μm) for all the specimens throughout the study.

**Intraoperative imaging procedures**. A secondary outcome of our study was to investigate different imaging systems. All standard of care surgical procedures had priority over study related procedures. Fluorescence imaging was performed at pre-defined time points: (i) just before the first incision; (ii) when tumor visibility was most optimal, as judged by the surgeon (e.g., after lump preparation for a lumpectomy); (iii) after excision of the surgical cavity to inspect for remaining fluorescent spots. If fluorescent spots were detected, the surgeon was allowed to biopsy these spots; (iv) on all the resection planes after excising the specimen to check for the presence of fluorescent spots. An intraoperative fluorescence margin assessment entails a combination of the fluorescence image of the surgical cavity and the fluorescence image of the excised specimen within 1 h after excision. The surgeon was able to look at a second monitor to evaluate the fluorescence images while performing the surgical procedure. During the imaging procedure, the ambient light in the surgical theater was switched off to prevent possible interaction with the fluorescence-imaging procedure itself. The standard of care was not influenced or altered by the imaging procedures. Regarding intrathoracal esophageal imaging, only extraluminal fluorescence imaging of the tumor and resection plane was performed.

In vivo TBR was calculated for all the HNSCC subjects whose mucosal tumors had been directly exposed to the intraoperative camera. We only calculated the in vivo TBRs for the HNSCC subjects because the other tumors (BC, EC, CRC) were deeper-seated and thus (a) not visible or (b) no reliable calculation could be done due to overlying tissue and differences in optical tissue properties. The mean fluorescence intensity (MFI) of both the tumor and background areas was calculated. The MFI was based on a region of interest (ROI) which was carefully determined based on a macroscopic examination. The TBR was calculated as tumor ROI (MFI tumor)/background ROI (MFI non-tumor tissue) per tissue slice.

The median TBR was calculated on a per subject base. The data (MFI, TBR) were plotted as graphs using GraphPad Prism, version 8.

**Specimen handling**. According to the SOC, the specimen was marked using sutures to aid orientation during tissue processing and cross-correlation with the histopathology immediately after excision. The exact location of the suture was tumor type and surgeon-dependent and was carefully documented to correlate fluorescence signals with tissue orientation.

**(Freshly) excised surgical specimen imaging procedures**. The specimens were stored, as much as possible, in the dark during all the tissue processing phases to prevent possible photobleaching of the imaging agent. All the surgical specimens were handled to conform to the standard of care which was not affected by study related procedures.

Immediately after excision all six resection planes of the specimen were imaged, size allowing (e.g., frontal, dorsal, lateral, medial, caudal, and cranial) by both the intraoperative camera system of choice as well as the PEARL-trilogy® system with a maximum duration of 60 min after surgical excision of the specimen. The combined imaging time of both devices was a maximum of 5 min. The specimens' resection planes were marked with blue and black ink. The use of other ink colors was avoided since these might interfere with the fluorescence in the NIR range. The restricted use of two colors of ink did not affect the standard of care for tissue processing by the pathologist but, if a third ink color was needed, green ink was used to define additional pathological resection margins of interest.

The fluorescence-imaging time points were adapted due to the SOC differences in specimen processing of the different tumor types. Briefly, the fresh BC specimens were sliced on the day of surgery and then formalin fixed whereas the other tumor types were sliced after formalin fixation of the whole resection specimen 1–3 days after surgery. The surgical specimen was serially sliced into ±0.5-cm-thick tissue slices. White light photographs were made during and directly after slicing for orientation purposes. After slicing, both sides of each tissue slice underwent fluorescence imaging in a light-tight environment (PEARL-trilogy®). The BC slices were therefore imaged ~120 min after excision. The other tumor types were imaged after formalin fixation and thus 1–3 days after excision.

Subsequently, a pathologist blinded for the recorded fluorescence images, examined the tissue slices macroscopically. This involved a gross examination by visual inspection and palpation alone and the selected regions of interest were embedded in paraffin blocks for further standard of care histological analysis. Other tissue samples that had not been selected by gross examination were embedded based on high fluorescent intensity spots or regions. In line with Koller et al., a standardized workflow was executed in order to cross-correlate the final histopathology results with the recorded fluorescence images of the tissue slices of interest[21]. Subsequently, hematoxylin/eosin (H/E) stained 4-μm sections were produced and analyzed by a board-certified pathologist who was also blinded for the obtained fluorescent imaging data.

**Quantification of ONM-100 fluorescence in tissue slices**. To further validate the intraoperative imaging feasibility as a primary outcome, the images of both sides of the respective tissue slices collected with PEARL-trilogy® were used to calculate the mean fluorescence intensity (MFI) of both tumor and background areas. The MFI was based on a Region of Interest (ROI) which was carefully determined on the H/E 4-μm slides by a board-certified pathologist blinded for the fluorescent imaging data and subsequently overlayed precisely on the corresponding tumor and normal tissue slices. Non-tumor tissue was considered to be any other tissue than the tumor tissue in the respective tissue slice. No distinction was made between specific non-tumor tissue types when defining the background. TBR was calculated as tumor ROI (MFI tumor)/background ROI (MFI non-tumor tissue) per tissue slice. A median TBR was calculated on a per subject base. The data (MFI, TBR) were plotted as graphs using GraphPad Prism, version 8.

**Specific ex vivo activation experiments**. An activation buffer (0.1 M sodium acetate-acetic acid) or phosphate-buffered saline (PBS) was added to human plasma with a certain concentration of ONM-100 and mixed with 20X PBS in a 96-well plate. The fluorescence was measured using a plate reader (TECAN Infinite M200 PRO, Männedorf, Switzerland).

The tumor was collected with adjacent stromal tissue during surgery and immediately frozen in optimal cutting temperature compound (OCT). It was cut into 8 μm frozen sections and sprayed with a fluorescent pH sensor (ICG was substituted with a tetramethylrhodamine (TMR) dye) and incubated for 3 min. After 10 min of fixation with formalin, the slides were washed three times with 0.9% NaCL + 0.5% Tween 20. The sections were DAPI stained and scanned for fluorescence (Zeiss Axio slide scanner, Oberkochen, Germany). Adjacent slides were H/E stained for histopathological correlation purposes.

**Statistical analysis**. The H/E sections, as shown in the respective figures, correspond to the tissue as shown in the adjacent panels. Note that from all included tissue slides, a corresponding H/E section has been cut and evaluated by the pathologist for correlating the fluorescence images to histolopathology. The MFI was calculated, using Image J Fiji, as total counts per ROI pixel area in both the tumor and background non-tumor tissues. The data was tested for Gaussian distribution using Anderson–Darling and Shapiro–Wilk tests; none of the data were normally distributed. Differences in fluorescence intensities between dose cohorts were tested using a Wilcoxon statistical test. The data are presented as median and interquartile ranges. A receiver-operator curve (ROC) was calculated using Graphpad Prism, version 8. A P value <0.05 was regarded as statistically significant. GraphPad Prism, version 8 was used for the statistical analyses.

**Reporting summary**. Further information on research design is available in the Nature Research Reporting Summary linked to this article.

## Data availability

All the data (imaging data, safety data, and pharmacokinetic data) gathered and/or processed during this study are available from the corresponding author on request. The source data (individual data points) underlying Figs. 3 and 5, Table 1 and Supplementary Figs. 1, 2, 4, and 6 are provided as a Source data file. All other data are available within the Article and Supplementary information.

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

## Acknowledgements

The authors want to thank all the participating subjects. Next, we would like to thank Gert-Jan Meersma, Nadine Baan and the Department of Pathology for their technical assistance, physician assistants Clara Lemstra, Arieke Prozee, Jacqueline Biermann, and Rachel Dopheide for their help in recruiting subjects, and Cristina Sala Ripoll for her help with the illustrations.

## Author contributions

F.J.V., P.J.S., M.K., B.D.S., and Y.J. designed the study. F.J.V. and P.J.S. performed the data acquisition, analyzed and interpreted the data and drafted the paper. T.Z. and J.G. designed the polymer composition of ONM-100. B.v.d.V. and J.J.D. were involved in the histopathological analyses and the reviewing of the paper. Members of the SHINE study group performed the fluorescence-imaging procedures, histopathological analyses and/or critically revised the paper. G.M.v.D. and M.J.H.W. designed and supervised the study, interpreted the data, and drafted the paper. All the authors reviewed the final paper. The research was funded by OncoNano Medicine Inc. J.G. and B.D.S. received National Institutes of Health funding (R01 CA192221, R01 CA211930).

## Competing interests

B.D.S. and J.G. are advisors, Y.J., T.Z., and Y.A. are employees and N.S. is a consultant at OncoNano Medicine Inc. B.D.S., J.G., Y.J., T.Z., Y.A., and N.S. are shareholders of OncoNano Medicine Inc. J.P.H. is an owner of JPH Clinical Development Inc and is a consultant at OncoNano Medicine Inc. The research was funded by OncoNano Medicine Inc. G.M.v.D. is a member of the Scientific Advisory Board of SurgVision BV as well as the CEO, founder, and shareholder of AxelaRx/TRACER BV. The sponsor (OncoNano Medicine Inc.) participated in the study design, the data were analyzed independently of the sponsor. The sponsor delivered technical advice for writing the paper (B.D.S. and J.G.). All other authors declare no competing interests.

## Additional information

## the SHINE study group

Y. Albaroodi[3], L. B. Been[2], F. Dijkstra[2], B. van Etten[2], Q. Feng[6], R. J. van Ginkel[2], K. Hall[3], K. Havenga[2], J. W. Haveman[2], P. H. J. Hemmer[2], L. Jansen[2], S. J. de Jongh[9], G. Kats-Ugurlu[4], W. Kelder[10], S. Kruijff[2], I. Kruithof[11], E. van Loo[2], J. L. N. Roodenburg[1], N. Shenoy[12], K. P. Schepman[1] & S. A. H. J. de Visscher[1]

[9]Department of Gastroenterology and Hepatology, University Medical Center Groningen, Groningen, The Netherlands. [10]Department of Surgery, Martini Hospital Groningen, Groningen, The Netherlands. [11]Department of Pathology, Martini Hospital Groningen, Groningen, The Netherlands. [12]Aravasc Inc., Sunnyvale, CA 94089, USA.

