## [Peer Review File · Nature Communications]

Reviewers' comments:

Reviewer #1 (Remarks to the Author): Expert in clinical trial and imaging

It was a pleasure to read this excellently designed paper.

It describes the new imaging device ONM-100.

The paper shows that ONM-100 is safe and has an enormous potential to visualize tumor margins, revolutionizing image guided surgery and becoming a game changer in surgery.

Main criticism:

1. unfortunately the authors did not fully recognize the high potential of ONM-100. Especially for problem localizations such as in the head and neck area ONM-100 will gain in importance in the future. By far the most common malignant tumors worldwide are basal cell carcinomas and squamous cell carcinomas (most common in the face). These tumors are often operated in two or more steps and very often have to be excised several times, because the tumor borders could not be recognized by the surgeon under macroscopic conditions. With the help of ONM-100 these tumors could possibly be operated on in one step in the future (excision of the tumor and immediate closure in one operation by exact imaging of the tumor), which could lead to an enormous cost reduction in the health care systems. Especially in the treatment of skin tumors in the facial area, the future medical but also the economic importance becomes apparent. Therefore, I would recommend to apply this new technique to some skin tumors as an example or at least to discuss its application to skin tumors in the facial area.

2. The false positive cases: Maybe these are no false positive cases at all? Maybe there are errors in histological diagnosis? Please explain in detail how histology was performed in the section Material and Methods. Please perform additional immunohistochemical staining or PCR etc. to definitely show that ONM-100 positive cases were really pathologically tumor-free.

Minor criticism:

Please discuss possible difficulties in using ONM-100 in areas where the pH is lower, such as the gastrointestinal tract. Could this also lead to false positive cases?

Reviewer #2 (Remarks to the Author): Expert in fluorescence imaging systems and cancer

In the manuscript "Nanoscale Macromolecular Cooperativity for pH Guided Clinical Therapy" Voskuil et al present a first-in-human study using a novel pH sensitive fluorescent contrast agent, ONM-100, in three different solid tumor types: head and neck squamous cell carcinoma (n=13), breast cancer (n=11), esophageal cancer (n=3), and colorectal cancer (n=3). The rationale for using a pH sensitive in tumor delineation during surgery is a more acidic extracellular environment compared with normal values resulting from aerobic glycolysis (Warburg effect). Such a phenomenon could highlight the presence of cancer using a freely, non-specific, circulating contrast agent and taking advantage of the Enhanced Permeability Retention effect in tumors. Importantly, the main advantage of such an approach would be to maximize tumor to background ratio since only the contrast agent located near the tumor would emit fluorescence. In this study, the pH sensitive contrast agent was injected intravenously to patients 24 hours prior to surgery and imaged using either open surgery or endoscopic fluorescence imaging systems first, followed by ex vivo imaging and pathology (H&E). Results are promising with no false negatives and additional metastases or secondary sites identified. However a few tumors (9 total) were not detected using this technology.

Overall, the study is novel, rigorous and well conducted. This reviewer notes impressive tumor-to-background ratio values and a promising strategy that has the potential to guide the surgeon during cancer resection. There are still a few comments & questions, but these do not diminish the

enthusiasm of this reviewer for the work presented in this manuscript.

1- The first comment relates to the title of the manuscript: "Nanoscale Macromolecular Cooperativity for pH Guided Clinical Therapy". The reviewer believes that this title does not describe accurately the content of the manuscript and should be changed to mention the name of the contrast agent, the fact that this is a first-in-human study and the application to cancer surgery. The current content does not provide insights on nanoscale macromolecular cooperativity, but rather illustrates the application of this process during oncologic surgery.

2- The authors implemented a smart workflow for the imaging all the way from surgery to specimen and pathology. This scheme allows to conveniently confirm the presence of cancer in the specimen. Unfortunately this reviewer is slightly disappointed that the figures mostly reports specimen imaging and only a few intraoperative pictures (4.g, 4.h, 4.r, 4.s in the main article and 5.b, 5.e, 5.h in the supplements). It would be appreciated to have more images of the surgeon perception since this is the ultimate goal of such method, i.e. to provide real-time tumor localization information. The movies are an interesting addition, but show the same cases as figure 4.

3- An important finding from this study is the limitation of near-infrared imaging fluorescence in detecting tumors that are deep into tissues. A total of 9 tumors were missed, which represents a significant amount, even though they were detected and removed with no positive margin with the standard procedure. However, this reviewer feels that this point should be discussed in more depth since it has profound implications on the potential use of this technology as a standard of care. Reasons why these tumors were not detected, the clinical implications and possibilities to improve detection of deep tumors should all be discussed. The last paragraph of the discussion insists on the potential clinical success but fails to include a discussion of the limitations such as illustrated in this study.

4- Another point of discussion should focus on the various conditions in which pH could be lower and identification of fluorescence with a pH sensitive contrast agent lead to false positive results. This reviewer thinks about cases of inflammation or oxygen deprivation.

5- Figure 4.s shows a hot spot at the bottom of the image (also visible when the movie is playing). What is the origin of this spot?

6- Several intraoperative imaging systems are used during this study. Because each system is fundamentally different, and therefore the performances and the fluorescence values reported change from one system to another, this reviewer believe that it brings confusion to this study. In other words, the values reported can change not because of the molecular process but because of the system used. This reviewers understands that the ultimate metric used in this study is tumor to background ratio. However, it would have been clearer to use a single open surgery and a single endoscopic system during this study. Adding more systems does not bring any value (at least in what the study reports in this manuscript).

7- Some details regarding the dose escalation values would be appreciated. For instance the MABEL, NOAEL and MTD values (when applicable) obtained from the preclinical studies would be interesting. This is especially true for the choice of the highest dose. Since the signal and TBR significantly increase with the dose, there seems to be an advantage in increasing the dose further, if possible.

8- The timing of injection is also an interesting point to discuss as already included in the manuscript. It might be interesting to detail why it was determined to inject patients 24h before surgery. Were the preclinical studies supporting this decision?

9- Can the authors include systematically all acquisition parameters (exposure time, gain) typically used for all imaging system? It is mentioned a standard exposure time of 100ms (i.e. a frame rate of 10 images per second) while using the Explorer Air. Is this a suitable frame rate for use during surgery? In addition, could the authors provide references to the imaging systems used?

10- It is mentioned that a calibration disk was used during the acquisition. Could the results of the sensitivity for each imaging system be included in supplementary information? This would certainly bring valuable information on the interpretation of the data to the readership.

11- It is stated that ambient light was turned off during fluorescence acquisitions. What this the case during the endoscopic procedures?

12- Figure 4 legend does not mention description for images 4.r and 4.s.

Response to referees

Original title: *Nanoscale Macromolecular Cooperativity for pH Guided Clinical Therapy*

Current title: *Exploiting metabolic acidosis in solid cancers using a tumor-agnostic novel pH-activatable nanoprobe for clinical fluorescence-guided surgery*

Tracking ID: NCOMMS-20-04645

Reviewers' comments:

Reviewer #1 (Remarks to the Author): Expert in clinical trial and imaging

It was a pleasure to read this excellently designed paper. It describes the new imaging device ONM-100. The paper shows that ONM-100 is safe and has an enormous potential to visualize tumor margins, revolutionizing image guided surgery and becoming a game changer in surgery.

Answer: We would like to thank the reviewer for the positive comments on the manuscript and acknowledging the potential for future implementation in surgery. We are more than willing to respond to the valuable comments provided below.

Main criticism:

1. Unfortunately the authors did not fully recognize the high potential of ONM-100. Especially for problem localizations such as in the head and neck area ONM-100 will gain in importance in the future. By far the most common malignant tumors worldwide are basal cell carcinomas and squamous cell carcinomas (most common in the face). These tumors are often operated in two or more steps and very often have to be excised several times, because the tumor borders could not be recognized by the surgeon under macroscopic conditions. With the help of ONM-100 these tumors could possibly be operated on in one step in the future (excision of the tumor and immediate closure in one operation by exact imaging of the tumor), which could lead to an enormous cost reduction in the health care systems. Especially in the treatment of skin tumors in the facial area, the future medical but also the economic importance becomes apparent. Therefore, I would recommend to apply this new technique to some skin tumors as an example or at least to discuss its application to skin tumors in the facial area.

Answer: We agree with the reviewer that the potential of ONM-100 goes beyond the four tumor types investigated in the current study. The reviewer is correct that one of the most important challenges, but also one of the largest, in surgical oncology is to facilitate the surgeon with real-time information on the extend of disease, thereby reducing the need for second (or third) surgeries to correct for inadequate therapy, i.e. positive margins. Indeed, since skin cancer is by far the most common cancer type worldwide, ONM-100 has the potential to reduce costs and patient burden for this category of patients significantly and fulfills a clinical unmet need. However, as skin cancer was not incorporated in the current

study protocol, and patient accrual has already closed, it is unfortunately not possible to include an additional cohort of patients with skin cancer at this stage. As a phase II study is currently ongoing, we are expanding the utility of ONM-100 (NCT03735680). In this study, Head and Neck Squamous Cell Carcinomas are (again) included in the design, therefore, we agree that skin cancer would be an excellent suggestion to include in future studies. Moreover, we added a statement to the discussion section regarding the potential for imaging of other cancer cancers. Line: 260-265

2. The false positive cases: Maybe these are no false positive cases at all? Maybe there are errors in histological diagnosis? Please explain in detail how histology was performed in the section Material and Methods. Please perform additional immunohistochemical staining or PCR etc. to definitely show that ONM-100 positive cases were really pathologically tumor-free.

Answer: We agree that we report five false positive cases, as also stated in the results and discussion section of the manuscript. We have rigorously analyzed these cases, since we agree with the reviewer that it is highly valuable to exclude all possibilities to erroneously report a false positive. Moreover, we are aware of the fact the potential sampling error during standard histopathological evaluation can occur. Despite these efforts, we cannot otherwise conclude that there was no tumor involvement in our four reported cases (see extensive explanation per case below) where there was involvement of premalignant dysplasia in one case. Although a high specificity with as limited false positives as possible is desired, we strongly believe that in surgical oncology a high sensitivity (i.e. to not miss true tumor lesions) is most important and thus inherently any diagnostic test will lead to false positives. In oncology the clinical impact of missing cancer (i.e. false negative) is much worse for the individual patient. In fact, we were able to visualize all tumors involved (i.e. 29) which was also the case for the margin assessment using ONM-100, since all 9 tumor-positive margins were detected using ONM-100 fluorescence.

Case 1 (patient 8): An intra-operative collected biopsy of the nerve tissue (n. lingualis), suspected for perineural growth based on ONM-100 fluorescence did not show any tumor tissue after extensive and detailed histopathological examination. We could argue that due to the size of the biopsy a sample error could have occurred, but this is only hypothetical.

Case 2 (patient 14) and Case 3 (patient 20): Extensive tissue analysis of these two cases showed fluorescence located around salivary glands, a phenomenon we and others observed previously in clinical trials investigating fluorescence imaging^{1,2}. We have performed detailed immunohistochemical staining on multiple tissue slides, however no tumor tissue could be detected in both cases.

Case 4 (patient 22): The tissue of this patient showed fluorescence located at the major pectoral muscle, a phenomenon observed in a previous clinical trial performed by our group with a different imaging strategy³ (i.e. antibody-fluorophore conjugate). The fluorescence was homogenously distributed at the fascia of muscle, and additional histopathological evaluation did not alter initial findings. We hypothesize that the fluorescence is present due to specific optical characteristics of the tissue itself, although we were not able to confirm this by extensive *ex vivo* imaging.

Special note:

Case 5 (patient 30): The excised whole specimen showed fluorescence located at the surgical margin which correlated to ductal carcinoma in situ (DCIS) at the resection margin, a pre-malignant condition, which can be therefore be debated as a true positive result. In this case, histopathology correlated with fluorescence imaging findings.

Minor criticism:

Please discuss possible difficulties in using ONM-100 in areas where the pH is lower, such as the gastrointestinal tract. Could this also lead to false positive cases?

Answer: We thank the reviewer for bringing up this interesting comment. We agree that, since ONM-100 responds to the acidic environment of solid tumors, the imaging agent can be activated in other disease areas and organ sites where the pH is low. Since ONM-100 is intravenously administered and only extravasates due to the enhanced permeability and retention (EPR) effect, in theory, it accumulates only in areas with semi-permeable vessels. Once extravasated, ONM-100 only fluoresces when the external pH (pH_e) is lowered due to the Warburg effect. Indeed, in theory a small amount of ONM-100 could reach the intraluminal space of the stomach for example in case of the presence of a bleeding or oozing tumor. Here, it could limit the discriminative strength between tumor and healthy surrounding tissue ones injected systemically. Although in theory this is possible, we did not notice fluorescence activation in the esophagus tissue, nor at the gastro-esophageal junction (GEJ). For future studies, it would indeed be highly relevant to include stomach carcinoma's, to see if this tumor type is also feasible to visualize with ONM-100 without aspecific activation by for instance a low pH of the intraluminal gastric fluids. We have further addressed this comment in the discussion section (line 236-242).

Reviewer #2 (Remarks to the Author): Expert in fluorescence imaging systems and cancer.

In the manuscript "Nanoscale Macromolecular Cooperativity for pH Guided Clinical Therapy" Voskuil et al. present a first-in-human study using a novel pH sensitive fluorescent contrast agent, ONM-100, in three different solid tumor types: head and neck squamous cell carcinoma (n=13), breast cancer (n=11), esophageal cancer (n=3), and colorectal cancer (n=3). The rationale for using a pH sensitive in tumor delineation during surgery is a more acidic extracellular environment compared with normal values resulting from aerobic glycolysis (Warburg effect). Such a phenomenon could highlight the presence of cancer using a freely, non-specific, circulating contrast agent and taking advantage of the Enhanced Permeability Retention effect in tumors. Importantly, the main advantage of such an approach would be to maximize tumor to background ratio since only the contrast agent located near the tumor would emit fluorescence. In this study, the pH sensitive contrast agent was injected intravenously to patients 24 hours prior to surgery and imaged using either open surgery or endoscopic fluorescence imaging systems first, followed by ex vivo imaging and pathology (H&E). Results are promising with no false negatives and additional metastases or secondary sites identified. However, a few tumors (9 total) were not detected using this technology.

Overall, the study is novel, rigorous and well conducted. This reviewer notes impressive

tumor-to-background ratio values and a promising strategy that has the potential to guide the surgeon during cancer resection. There are still a few comments & questions, but these do not diminish the enthusiasm of this reviewer for the work presented in this manuscript.

Answer: We would like to thank the reviewer for the very positive comments on the manuscript and the conduction of the study. We are more than willing to respond to the comments and questions raised by the reviewer. We first would like to respond to the following statement: *However, a few tumors (9 total) were not detected using this technology.* Unfortunately, we think the reviewer misinterpreted some of the data. In the current study, we have included 30 patients, as the reviewer stated correctly. Of these thirty patients, 29 patients were applicable for further analysis, since one patient with esophageal cancer (EC) did not have viable tumor cells after neo-adjuvant chemoradiation therapy in the specimen as confirmed by final histopathology. Important to state: All 29 tumors were detected using ONM-100, see also Supplemental Figure 6.

Of these remaining 29 patients, two patients only underwent removal of small tissue biopsies because of the suspected presence of peritoneal metastasis (2 colorectal cancer (CRC) patients). Although, ONM-100 did detect the presence of tumor tissue in these biopsies (see Figure Supplemental Figure 5b and Supplemental Figure 6), the fact that the primary CRC tumor was not excised (and therefore no availability of the specimen) as decided by the attending surgeon. The decision was based on the lack of clinical value for the patient in terms of limited prognosis and reduced quality of life, precluded the possibility for margin assessment. Of note, for diagnostic purposes of ONM-100 like in peritoneal carcinomatosis, the goal is not to achieve a free margin, but to collect reliable tissue for the assessment of the presence of cancer cells which has an impact on the remaining surgical procedure. The two remaining EC patients and one remaining CRC patient all had a tumor-negative margin based on ONM-100 fluorescence, which was in line with final histopathology which reported a tumor-negative margin for all three patients. Since the very small number of patients in both tumor types, and the lack of tissue to compare as a tumor-positive margin, we excluded these patients for margin assessment metrics.

In total, we did use 24 excised tissues (head and neck squamous cell carcinomas and breast cancer specimens) to correlate ONM-100 fluorescence results for margin analyses and final histopathology of margin status. Of these 24 patients, 9 patients were diagnosed with a histological tumor-positive surgical margin. In fact, ONM-100 detected all tumor positive margins within one hour after excision, which underlines the potential added value for the attending surgeon.

Although not specifically specified, it might be the fact that the reviewer is referring to the following sentence of the results section: *Thirteen HNSCC, the five superficially seated BC tumors and two CRC tumors were visualized in-situ in real time by ONM-100 fluorescence during surgery.* Indeed, a number of 9 tumors was not visualized, but this is only the case for *in vivo* imaging. In fact, this can be considered as desirable, since the penetration depth of near-infrared red is up to several millimeters. So, one could argue that if no fluorescence is visible during surgery, a margin of at least a couple of millimeters is present. Consequently, a surgeon performing a radical excision with a sufficient margin of overlaying healthy tissue, limiting penetration of emitted NIR fluorescent light, actually does not want to detect any

fluorescence, as presence of fluorescence implies a positive margin. The fact that all tumors were visualized *ex vivo* after specimen bread loaf slicing, also stated previously, confirms the reliability of the mechanism of ONM-100 for intraoperative imaging.

1- The first comment relates to the title of the manuscript: “*Nanoscale Macromolecular Cooperativity for pH Guided Clinical Therapy*”. The reviewer believes that this title does not describe accurately the content of the manuscript and should be changed to mention the name of the contrast agent, the fact that this is a first-in-human study and the application to cancer surgery. The current content does not provide insights on nanoscale macromolecular cooperativity, but rather illustrates the application of this process during oncologic surgery.

Answer: We would like to thank the reviewer for addressing this point of concern. We agree that a different title, thereby providing better insight on the content of the manuscript, would better suit the manuscript. Therefore, we have adjusted the manuscript title to: ‘*Exploiting metabolic acidosis in solid cancers using a tumor-agnostic novel pH-activatable nanoprobe for clinical fluorescence-guided surgery*’.

2- The authors implemented a smart workflow for the imaging all the way from surgery to specimen and pathology. This scheme allows to conveniently confirm the presence of cancer in the specimen. Unfortunately, this reviewer is slightly disappointed that the figures mostly reports specimen imaging and only a few intra-operative pictures (4.g, 4.h, 4.r, 4.s in the main article and 5.b, 5.e, 5.h in the supplements). It would be appreciated to have more images of the surgeon perception since this is the ultimate goal of such method, i.e. to provide real-time tumor localization information. The movies are an interesting addition, but show the same cases as figure 4.

Answer: We would like to thank the reviewer for his comments on the workflow for translating *in vivo* and *ex vivo* fluorescence results to histopathological confirmation. We agree that the majority of figures is focused on the (freshly) excised specimen, since we believe that the *ex vivo* environment provides a more reliable image acquisition since factors such as distance of imaging, angle of imaging, homogenous illumination and environmental light can be controlled in a more standardized method^{2,4-6}. Nevertheless, we agree with the reviewer that a major part of the clinical decision making should and can be done prior to excision (i.e. *where to cut*). *In vivo* imaging can assist in delineating the tumor for this purpose. Therefore, we have added additional *in vivo* images of the surgeon perception as supplemental figure to the manuscript (see Supplemental Fig. 8 and Movie M4). We hope the reviewer appreciates the additional figures and video provided.

3- An important finding from this study is the limitation of near-infrared imaging fluorescence in detecting tumors that are deep into tissues. A total of 9 tumors were missed, which represents a significant amount, even though they were detected and removed with no positive margin with the standard procedure. However, this reviewer feels that this point should be discussed in more depth since it has profound implications on the potential use of this technology as a standard of care. Reasons why these tumors were not detected, the clinical implications and possibilities to improve detection of deep tumors should all be

discussed. The last paragraph of the discussion insists on the potential clinical success but fails to include a discussion of the limitations such as illustrated in this study.

Answer: For this comment, we would like to refer to our initial response to this reviewer (see page 4 of this document). In short, we believe the reviewer has misinterpreted some of the data, since ONM-100 did visualize all 29 tumors (one EC patient did not have viable tumor cells at the time of surgery due to neo-adjuvant chemoradiation).

4- Another point of discussion should focus on the various conditions in which pH could be lower and identification of fluorescence with a pH sensitive contrast agent lead to false positive results. This reviewer thinks about cases of inflammation or oxygen deprivation.

Answer: We fully agree with the reviewer that this is an important point to address. It is known that within inflammatory tissue lower pH levels are present based on mechanisms like opsonization, although the buffering capacity of surrounding tissue seems to counteract a possible lowering in pH in these conditions. Similarly, this is the case in acute and more chronic ischemia or hypoxia of healthy tissue such as muscle or organs leading to a systemic lowering of pH, as if often the case in sepsis. The co-existence of a low systemic pH in patients with solid tumors is uncommon. Solid tumors do experience relative chronic hypoxia and the phenomenon is actually one of the underlying factors leading to a changed anaerobic metabolism resulting in a decreased intratumoral pH. Taking the above into consideration, a false-positive due to inflammation might be more likely as some tumors do experience of peri-tumoral inflammatory rim as part of the defense mechanism of the innate immune system to the invading tumor cells. If this is the case, which we did not observe in this small series, it actually might result in a false-positive result but with minimal impact as the tumor margin is visualized by the lower pH caused by peri-tumoral inflammation. We have elaborated on this inflammatory mechanism leading to a potential false-positive result in the discussion and again thank the reviewer for his valuable comment (line 236-242).

5- Figure 4.s shows a hot spot at the bottom of the image (also visible when the movie is playing). What is the origin of this spot?

Answer: This image indeed shows an additional hot spot at the bottom of the image. This hot spot can either be reflectance, although this is limited in the NIR region, or more likely melanin pigmentation in the areola of the breast, which autofluoresces in the 800 nm range as described by Han et al ⁷.

Figure 1 | Autofluorescence of cutaneous melanins⁷.

Moreover, as this additional spot is not in the surgical field for tumor surgical resection and unlikely to be associated with the primary tumor process, this fluorescence activation of skin tissue is not clinically relevant for surgical decision making and can be assigned as background. As with any imaging technique, image interpretation is always within a context of anatomical region and biological processes, similar to X-ray imaging, nuclear and MRI imaging. Novel image analytical techniques like Artificial Intelligence and Deep Learning will provide tools to enhance image interpretation data in the near future for fluorescence-guided surgery.

6- Several intraoperative imaging systems are used during this study. Because each system is fundamentally different, and therefore the performances and the fluorescence values reported change from one system to another, this reviewer believe that it brings confusion to this study. In other words, the values reported can change not because of the molecular process but because of the system used. This reviewer understands that the ultimate metric used in this study is tumor to background ratio. However, it would have been clearer to use a single open surgery and a single endoscopic system during this study. Adding more systems does not bring any value (at least in what the study reports in this manuscript).

Answer: We agree with the reviewer that not only the concentration of the fluorophore contributes to the obtained fluorescence intensity values, both also a variety of external factors such as distance of imaging, angle of imaging, homogenous illumination and environmental light and the spatial resolution and sensitivity of the imaging device⁴. Therefore, we believe that reporting on the fluorescence intensity values obtained by the *ex vivo* imaging conditions can be more standardized and are therefore more reliable for comparison between tissue specimens. Important to state is that for **all** *ex vivo* measurements, only one type of imaging system is used (PEARL Trilogy, Li-Cor Biosciences). In fact, we fully agree with the reviewer that focusing on the tumor-to-background ratio (TBR) can (partly) correct for this, since the TBR is using its own background value as internal control. Nevertheless, we believe that it is a valuable asset to report on the ability of ONM-100 to be

detectable with a variety of (*in vivo*) imaging systems as in clinical practice not all surgeons all have the same camera systems due to different constraints on the hospital budget, reimbursement strategies etc. Evaluating ONM-100 by different camera systems increases the chance of faster adaptation in the field of surgical oncology. Moreover, due to the tumor-agnostic character of ONM-100, imaging of different cancers and thus locations of cancers might necessitate different imaging system (e.g. open versus laparoscopic and robotic surgery). Nevertheless, in order to compare results in larger clinical trials using ONM-100 with different camera systems, there has been a recent significant international collaborative effort in developing tissue-simulating solid phantoms which makes it possible to calibrate and validate camera systems in-between imaging sessions for quality control reasons, but also in-between different camera systems among different centers (See Figure 2 below) ⁸. In the reported study using ONM-100, we used a prototype calibrations disk as these phantoms were not available at the time of conducting the study (Figure 3).

Fig. 1. Concentrations of various phantom constituents

Figure 2 | Composite Phantom for benchmarking fluorescence imaging devices ⁸.

7- Some details regarding the dose escalation values would be appreciated. For instance, the MABEL, NOAEL and MTD values (when applicable) obtained from the preclinical studies would be interesting. This is especially true for the choice of the highest dose. Since the signal and TBR significantly increase with the dose, there seems to be an advantage in increasing the dose further, if possible.

Answer: We agree that these details are important, especially when designing a phase I first in human study. Due to the clinical nature of the current manuscript, we decided to exclude them from the manuscript. Of course, we are willing to provide the NOAEL here to be as transparent as possible.

Regarding the statement of increasing the dose further to increase the TBR further, we did notice this positive increase in TBR for the higher doses as well. Therefore, in the design of a subsequent phase II study, we will investigate higher doses as well (NCT03735680). This is now stated more explicitly in the discussion section. Line 262-263.

NOAEL: The no observed adverse effect level (NOAEL) for rats and dogs is the limit dose of 30 mg/kg of ONM-100 in a single two-minute intravenous bolus. The dog is considered the most sensitive species, based on liver alterations at the high dose of 30 mg/kg.

8- The timing of injection is also an interesting point to discuss as already included in the manuscript. It might be interesting to detail why it was determined to inject patients 24h before surgery. Were the preclinical studies supporting this decision?

Answer: We agree that the timing of the administration is an important factor in the design of the study and consequently, a factor which might contribute to the obtained data. As this was a first in-human study, with already several parameters investigated (e.g. different tumor types, different images devices), we choose not to include other parameters which would limit comparability between patients. Moreover, the chosen timepoint of injection was based on a pre-clinical study in rodents, in which the optimal administration to imaging time ranged from 6 to 24 hours^{9,10}. Due to practical considerations, 24 hours was considered ideal due to hospital logistics. This is also discussed in line 260-261.

9- Can the authors include systematically all acquisition parameters (exposure time, gain) typically used for all imaging system? It is mentioned a standard exposure time of 100ms (i.e. a frame rate of 10 images per second) while using the Explorer Air. Is this a suitable frame rate for use during surgery? In addition, could the authors provide references to the imaging systems used?

Answer: We agree that acquisition parameters are important variables for imaging devices. We have chosen to use imaging systems which are widely used in the field of fluorescence imaging with a world-wide large install-base like Novadaq/Stryker, Olympus and Intuitive Surgical, since this would allow rapid implementation in future larger phase II/III trials and eventually in the clinic, since no custom-made camera devices are necessary. The SurgVision Explorer Air® is widely used in fluorescence-guided surgery studies (see refs^{2,3,11}), as well as the Novadaq Spy Elite (CE-marked) (see ref¹²). For the endoscopic imaging, we have used the

CE-marked add-ons of existing endoscopic system, namely the Olympus NIR laparoscopy (see refs^{13,14}) system and the Intuitive Surgical daVinci Firefly laparoscope (see ref¹⁵). In the current study, we observed that a frame rate of 10 images per second is sufficient for intra-operative fluorescence guidance (Explorer Air), although we expect that in the not too far future, with increasing sensitivity of imaging devices this would allow an even higher frame rate. For the Novadaq Spy Elite, we used 7.5 frames per second. This is stated more explicitly in the manuscript (line 497-498). Both frame rates allowed for adequate real-time tumor detection using ONM-100.

10- It is mentioned that a calibration disk was used during the acquisition. Could the results of the sensitivity for each imaging system be included in supplementary information? This would certainly bring valuable information on the interpretation of the data to the readership.

Answer: We have used a custom-build calibration disk for the comparison between imaging devices. We have added the results of this imaging experiment in this response document, since we believe the focus of the manuscript should be on the clinical data.

Figure 3 | Comparison of different imaging devices using a custom build calibration disk

11- It is stated that ambient light was turned off during fluorescence acquisitions. Was this the case during the endoscopic procedures?

Answer: We did turn of the white light during the endoscopic procedures and only used NIR light at the time of imaging. With the current laparoscopic systems surgeons can switch between the imaging modalities by a button.

12- Figure 4 legend does not mention description for images 4.r and 4.s.

Answer: We have corrected the legend of Figure 4.

References

1. de Boer, E. *et al.* In Vivo Fluorescence Immunohistochemistry: Localization of Fluorescently Labeled Cetuximab in Squamous Cell Carcinomas. *Sci Rep* **5**, 14 (2015).
2. Voskuil, F. J. *et al.* Fluorescence-guided imaging for resection margin evaluation in head and neck cancer patients using cetuximab-800CW: A quantitative dose-escalation study. *Theranostics* **10**, 3994–4005 (2020).
3. Koller, M. *et al.* Implementation and benchmarking of a novel analytical framework to clinically evaluate tumor-specific fluorescent tracers. *Nat Commun* **9**, 37–39 (2018).
4. Koch, M., Symvoulidis, P. & Ntziachristos, V. Tackling standardization in fluorescence molecular imaging. *Nature Photonics* **12**, 505–515 (2018).
5. van Keulen, S. *et al.* Rapid, non-invasive fluorescence margin assessment: Optical specimen mapping in oral squamous cell carcinoma. *Oral Oncol* **88**, 58–65 (2019).
6. van Keulen, S. *et al.* The Sentinel Margin: Intraoperative Ex Vivo Specimen Mapping Using Relative Fluorescence Intensity. *Clin. Cancer Res.* **25**, 4656–4662 (2019).
7. Han, X., Lui, H., McLean, D. I. & Zeng, H. Near-infrared autofluorescence imaging of cutaneous melanins and human skin in vivo. *J Biomed Opt* **14**, 024017 (2009).
8. Gorpas, D. *et al.* Multi-Parametric Standardization of Fluorescence Imaging Systems Based on a Composite Phantom. *IEEE Trans Biomed Eng* **67**, 185–192 (2020).
9. Zhao, T. *et al.* A transistor-like pH nanoprobe for tumour detection and image-guided surgery. *Nat Biomed Eng* **1**, 1–8 (2016).
10. Wang, C. *et al.* Optical molecular imaging for tumor detection and image-guided surgery. *Biomaterials* **157**, 62–75 (2018).
11. Gao, R. W. *et al.* Determination of Tumor Margins with Surgical Specimen Mapping Using Near-Infrared Fluorescence. *Cancer Res* **78**, 5144–5154 (2018).
12. Rother, U. *et al.* Dosing of indocyanine green for intraoperative laser fluorescence angiography in kidney transplantation. *Microcirculation* **24**, (2017).
13. van Dam, D. A. *et al.* Comparing Near-Infrared Imaging with Indocyanine Green to Conventional Imaging During Laparoscopic Cholecystectomy: A Prospective Crossover Study. *J Laparoendosc Adv Surg Tech A* **25**, 486–492 (2015).
14. De Neef, A. *et al.* Fluorescence of Deep Infiltrating Endometriosis During Laparoscopic Surgery: A Preliminary Report on 6 Cases. *Surg Innov* **25**, 450–454 (2018).
15. Spinoglio, G., Bertani, E., Borin, S., Piccioli, A. & Petz, W. Green indocyanine fluorescence in robotic abdominal surgery. *Updates Surg* **70**, 375–379 (2018).

REVIEWERS' COMMENTS:

Reviewer #1 (Remarks to the Author):

In my view, the authors answered all comments and questions from the reviewers more than satisfactorily! I firmly believe that this new innovative nanoprobe has the potential to revolutionize image guided surgery.

Reviewer #2 (Remarks to the Author):

Following the previous version, all this reviewer's comments were appropriately addressed and all these improvements have greatly improved the quality of this manuscript.

This reviewer has no further comments to make on this manuscript and would like to congratulate the authors for the quality of their work.

Final revision responses

Title: *Exploiting metabolic acidosis in solid cancers using a tumor-agnostic novel pH-activatable nanoprobe for clinical fluorescence-guided surgery*

Tracking ID: NCOMMS-20-04645A

Reviewers' comments:

Reviewer #1 (Remarks to the Author):

In my view, the authors answered all comments and questions from the reviewers more than satisfactorily! I firmly believe that this new innovative nanoprobe has the potential to revolutionize image guided surgery.

Answer: We would like to thank the reviewer for the smooth reviewing process and excellent feedback.

Reviewer #2 (Remarks to the Author):

Following the previous version, all this reviewer's comments were appropriately addressed and all these improvements have greatly improved the quality of this manuscript.

This reviewer has no further comments to make on this manuscript and would like to congratulate the authors for the quality of their work.

Answer: We would like to thank the reviewer for the smooth reviewing process and excellent feedback.